# Strain-Enhanced Thermoelectric Performance in GeS_2_ Monolayer

**DOI:** 10.3390/ma15114016

**Published:** 2022-06-06

**Authors:** Xinying Ruan, Rui Xiong, Zhou Cui, Cuilian Wen, Jiang-Jiang Ma, Bao-Tian Wang, Baisheng Sa

**Affiliations:** 1Key Laboratory of Eco-Materials Advanced Technology, College of Materials Science and Engineering, Fuzhou University, Fuzhou 350100, China; ruanxinyingjg@163.com (X.R.); rxiong421@163.com (R.X.); cuizhoufzu@163.com (Z.C.); clwen@fzu.edu.cn (C.W.); 2Institute of High Energy Physics, Chinese Academy of Sciences (CAS), Beijing 100049, China; majj88@ihep.ac.cn; 3Spallation Neutron Source Science Center (SNSSC), Dongguan 523803, China; 4Collaborative Innovation Center of Extreme Optics, Shanxi University, Taiyuan 030006, China

**Keywords:** GeS_2_ monolayer, strain engineering, first-principles calculations, thermoelectric materials, thermal conductivity

## Abstract

Strain engineering has attracted extensive attention as a valid method to tune the physical and chemical properties of two-dimensional (2D) materials. Here, based on first-principles calculations and by solving the semi-classical Boltzmann transport equation, we reveal that the tensile strain can efficiently enhance the thermoelectric properties of the GeS_2_ monolayer. It is highlighted that the GeS_2_ monolayer has a suitable band gap of 1.50 eV to overcome the bipolar conduction effects in materials and can even maintain high stability under a 6% tensile strain. Interestingly, the band degeneracy in the GeS_2_ monolayer can be effectually regulated through strain, thus improving the power factor. Moreover, the lattice thermal conductivity can be reduced from 3.89 to 0.48 W/mK at room temperature under 6% strain. More importantly, the optimal ZT value for the GeS_2_ monolayer under 6% strain can reach 0.74 at room temperature and 0.92 at 700 K, which is twice its strain-free form. Our findings provide an exciting insight into regulating the thermoelectric performance of the GeS_2_ monolayer by strain engineering.

## 1. Introduction

Thermoelectric technology is one of the most fantastic energy-conversion technologies that can convert heat energy and electrical energy into each other directly [1,2,3]. Thermoelectric materials have recently gained extensive attention as a critical factor for thermoelectric technology. The figure of merit ZT can be directly used to visualize the thermoelectric conversion efficiency of thermoelectric materials and can be calculated by [4,5,6,7]:(1)$$\mathrm{ZT}=\frac{S^{2}\sigma T}{\kappa }$$
where *S* stands for the Seebeck coefficient, *σ* is electrical conductivity, and *T* represents temperature. κ is the thermal conductivity, consisting of both electronic and lattice parts. Herein, the thermoelectric power factor (PF) can be defined as PF = *S*^2^*σ*. Apparently, a higher PF and lower *κ* can contribute to an immense ZT value.

The development of 2D materials provides an excellent platform for discovering novel high-performance thermoelectric materials [8,9,10,11,12,13]. Previous studies have reported graphene [14,15], phosphorene (BP) [16,17,18], IVA–VIA compounds [19,20,21], and transition metal dichalcogenides (TMDs) [22,23,24], and all show excellent thermoelectric performance. In particular, IVA–VIA compounds exhibit high ZT values due to their ultralow lattice thermal conductivities [19,20]. Recently, the 1T-GeS_2_ monolayer has been reported as a potential thermoelectric material due to its relatively high electronic fitness function (EFF) value from high-through computational screening [21]. Moreover, the high-power factor of the GeS_2_ monolayer further reveals its great potential application in the field of thermoelectrics [25]. However, the ZT value of the 1T-GeS_2_ monolayer is only 0.23 when the thermal transport property is considered [25], which significantly hinders its further application. Therefore, it is of great significance to improve its thermoelectric performance by adjusting the thermal transport properties of GeS_2_ monolayers. It is worth mentioning that the electronic structures of 2D materials are easily affected by applied strains [26,27,28]. Strain engineering has been theoretically and experimentally proposed as a valid way to enhance the thermoelectric properties of 2D thermoelectric materials [29,30]. Experimentally, the thermal conductivity of the Bi_2_Te_3_ monolayer can be reduced by 50% by applying a tensile strain of 6% [31]. Theoretically, tensile strain can significantly enhance Seebeck coefficients while reducing thermal conductivity, and this has been observed in the PtSe_2_ monolayer [32]. Therefore, it is very interesting to investigate the strain effect on the electronic and thermoelectric properties of the GeS_2_ monolayer.

In the present work, based on first-principles calculations and by solving the semi-classical Boltzmann transport equation, we systematically studied the tensile strain effects on the thermoelectric properties of the GeS_2_ monolayer, including electronic structures, electronic transport properties, and phonon transport properties. It was found that the valence band near the Fermi level of the GeS_2_ monolayer will degenerate under tensile strain, which leads to an improvement in the power factor. Meanwhile, the phonon group velocities and phonon relaxation times decrease with an increasing tensile strain, resulting in a reduction in the lattice thermal conductivity, thereby enhancing the thermoelectric performance. Our results provided a new tactic for improving the thermoelectric properties of the GeS_2_ monolayer.

## 2. Methods

Our simulation works were based on first-principles calculations with the projector augmented-wave (PAW) [33] method, which is executed by the VASP [34] code, and the corresponding results were dealt with the ALKEMIE platform [35]. The generalized gradient approximation [36] with the Perdew–Burke–Ernzerhof functional (GGA-PBE) [37] was used to deal with the interaction between electronics and ions. The structure of the GeS_2_ monolayer was completely optimized until the energy and force convergence criteria were less than 10^−6^ eV and −0.01 eV, respectively. The cutoff energy was set to 600 eV, and a *k*-point mesh of 15 × 15 × 1 was adopted [38]. A vacuum thickness of 20 Å perpendicular to the in-plane direction of the GeS_2_ monolayer was built. The Heyd–Scuseria–Ernzerhof (HSE06) [39] hybrid functional with a range-separation parameter of 0.2 and mixing parameter of 0.25 was also adopted to obtain more accurate band structures and electronic transport properties of the GeS_2_ monolayer. The ab initio molecular dynamics (AIMD) simulations with the Nosé –Hoover thermostat (NVT) ensemble and a time step of 2ps were performed to investigate the thermal stability of the GeS_2_ monolayer [40,41]. 

A denser k-point mesh of 35 × 35 × 1 was used for static calculations to obtain more accurate electronic structures to solve semi-classical Boltzmann transport equations, which is realized in the BoltzTraP code [42]. The phonon spectrum and second-order anharmonic force constants were calculated by the Phonopy package [43] with a 6 × 6 × 1 supercell, while a 4 × 4 × 1 supercell was used to calculate third-order interatomic force constants. The sixth nearest neighbors were selected to obtain the third-order interatomic force constants to ensure the accuracy of lattice thermal conductivity and save the calculation time. Combing with second-order anharmonic force constants and third-order interatomic force constants as input files, the lattice thermal conductivity of the GeS_2_ monolayer can be obtained through the ShengBTE code [44].

## 3. Results and Discussion

### 3.1. Structural Stability and Band Structure

Similar to the 1T-MoS_2_ monolayer [45], each unit cell of the GeS_2_ monolayer consists of one Ge atom and two S atoms with the Ge sublayer sandwiched between two S sublayers. The side and top views of the GeS_2_ monolayer are plotted in Figure 1a,b, respectively. The relaxed lattice parameters are *a* = *b* = 3.44 Å, which agree with previous theoretical predictions [21,25]. Figure 1c describes the atom orbitals project band structure of the GeS_2_ monolayer. It is clear that the GeS_2_ monolayer demonstrates indirect band gap semiconductor features with a band gap of 1.50 eV. It is noted that the relatively large band gap can effectively prevent the bipolar conduction behavior in the materials and thus prevents the thermoelectric performance from being destroyed. Moreover, the VBM is mainly contributed by the S-*p* orbital, while the CBM is occupied by both Ge-*s* and S-*p* orbitals. Our results are in accordance with the previous theoretical predicated [25,46], indicating that our calculation parameters are reasonable.

To understand the stability of the GeS_2_ monolayer, we then conducted phonon spectrum calculations and AIMD simulations to explore the lattice and thermal dynamic stabilities, respectively. Figure 2a describes the phonon spectrum for the GeS_2_ monolayer. Obviously, there are nine dispersion curves with three acoustic branches and six optical branches since a GeS_2_ unit cell contains three atoms. Moreover, no imaginary frequency can be found in phonon dispersion curves, indicating that the GeS_2_ monolayer possesses a good lattice dynamic stability. It is noted that the ZA mode for the GeS_2_ monolayer near the Γ point is quadratically converged, which can be usually observed in 2D materials systems [47]. Furthermore, from the PhDOS of the GeS_2_ monolayer, we know that the low- and high-frequency regions are mainly contributed by Ge and S atoms, respectively. Moreover, the phonon spectrum of the GeS_2_ monolayer under 2% compressive strain was also calculated, as shown in Appendix A. A negative frequency was observed in the phonon spectrum, indicating the instability of the GeS_2_ monolayer under compressive strain. Hence, in our study, we mainly concentrated on the tensile strain effects on the thermoelectric properties of the GeS_2_ monolayer. Figure 2b illustrates the energy evolution and structure snapshot of the GeS_2_ monolayer for 10 ps at 300 K. It is clear that the changes in total energy are minimal, and atoms are slightly vibrating around their equilibrium positions, suggesting that the GeS_2_ monolayer exhibits excellent thermal dynamic stability as well. 

Figure 3 illustrates the band structures of the GeS_2_ monolayer at different biaxial tensile strains. Herein, the tensile strains can be calculated by ε = (*a* − *a*_0_)/*a*_0_ × 100%, where *a*_0_ stands for the lattice constant when unstrained, while *a* represents the lattice constant under strain. Obviously, within our investigated strain range (0~6%), the band gap of the GeS_2_ monolayer increases gradually with tensile strain since CBM moves toward the higher energy level. Additionally, with the increases in tensile strain, the valence bands between K and Γ points move toward the Fermi level, which can enhance the degeneracy of the valence band and thus improve the Seebeck coefficient. Moreover, the band structure of the GeS_2_ monolayer under 8% tensile strain was also calculated, as shown in Appendix A. However, the valence band maximum shifts to the position between Γ and K under 8% tensile strain. This phenomenon will decrease band degeneracy in the GeS_2_ monolayer, which is not conducive to the thermoelectric application. Hence, in our study, we mainly concentrate on the 2–6% tensile strain effects on the thermoelectric properties of the GeS_2_ monolayer. These consequences indicate that the tensile strain can effectively regulate the electronic structures of the GeS_2_ monolayer. Therefore, an improvement in thermoelectric performance in the GeS_2_ monolayer is anticipated [48,49].

### 3.2. Electronic Transport Properties

We next investigate the effect of biaxial tensile strains on the electronic transport properties of the GeS_2_ monolayer, including the Seebeck coefficient (*S*), electric conductivity (*σ*), electronic thermal conductivity (*κ*_e_), and the power factor (PF). Figure 4 shows the contour maps of the Seebeck coefficient with respect to chemical potential under different biaxial tensile strains. Clearly, the *S* increases with an increasing tensile strain and decreases with an increasing temperature. The maximum *S* increases from 2386 μVK^−1^ (2318 μVK^−1^) to 2697 μVK^−1^ (2605 μVK^−1^) under p-type (n-type) doping, as the tensile strain augments from 0 to 6%. This phenomenon is mainly contributed by enlarging the band gap and band degeneracy in the GeS_2_ monolayer with the increase in tensile strain.

On the other hand, Figure 5a–d shows the electrical conductivity divided by the relaxation time (*σ*/*τ*) of the GeS_2_ monolayer under different tensile strains. Contrary to the Seebeck coefficients, electrical conductivity is insensitive to the temperature and decreases with an increasing tensile strain. A similar tendency as *σ*/*τ* can be observed in electronic thermal conductivity (Figure 6a–d) since it can be calculated by [50]: *κ*_e_ = L*σT*, where L represents the Lorenz number. Our results above show that the *S* and *σ*/*τ* exhibit opposite trends under tensile strain. Hence, we also calculated the power factor (PF) under different tensile strains, and the corresponding results are shown in Figure 7a–d. Apparently, the optimal value of the PF under p-type doping is much higher than n-type doping for all cases. More importantly, the PF gradually increases as the tensile strain is applied, which is due to the fact that the applied tensile strain has a more significant effect on the *S* than the *σ*/*τ*. The power factor as a function of carrier concentrations is also plotted in Appendix A.

### 3.3. Phonon Dispersion Curves and Transport Properties

Phonon thermal transport property is another critical factor for thermoelectric materials. Hence, the effect of tensile strain on the phonon transport properties of the GeS_2_ monolayer was investigated in the following. The phonon dispersion curves under different strains are illustrated in Figure 8. Clearly, no negative frequency was observed in any of the cases, suggesting that the GeS_2_ monolayer’s lattice is dynamically stable under these tensile strains. Furthermore, the frequencies of both optical and acoustic phonon modes gradually decrease with the increase in the tensile strain, leading to reducing phonon group velocities and thus a lower lattice thermal conductivity. This phenomenon is beneficial for the application of GeS_2_ monolayer in the fields of thermoelectrics.

To evaluate the convergence of the lattice’s thermal conductivity, we calculated the lattice thermal conductivity as a function of the nearest neighbor atomic, which is plotted in Appendix A. It is noted that the lattice thermal conductivity can reach good convergence criteria when the nearest neighbor atom is up to six. Figure 9a describes the lattice thermal conductivity ($\kappa_{l}$) of the GeS_2_ monolayer with respect to temperature under different tensile strains. It is interesting to note that $\kappa_{l}$ decreases with both increasing temperature and tensile strain. For example, the $\kappa_{l}$ of the unstrained GeS_2_ monolayer reduces from 3.89 to 1.13 W/mK when the temperature increases from 300 K to 1000 K. More importantly, the $\kappa_{l}$ will reduce to 0.48 W/mK when 6% strain is applied at 300 K. Such a small $\kappa_{l}$ is comparable to some recently reported novel 2D thermoelectric materials, such as a SnTe monolayer (0.67 W m^−1^ K^−1^) [51], Sb_2_Te_2_Se monolayer (0.46 W m^−1^ K^−1^) [52], and HfSe_2_ monolayer (0.7 W m^−1^ K^−1^) [53]. To unravel the strain-induced reduced lattice thermal conductivity behavior in the GeS_2_ monolayer, we also calculated the phonon group velocities ($\nu_{\lambda }^{}$) and phonon relaxation times ($\tau_{\lambda }$) since $\kappa_{l}$ can be obtained by [54]:(2)$$\kappa_{l}=\frac{\sum_{\lambda }C_{\lambda }\nu_{\lambda }^{2}\tau_{\lambda }}{V}$$
where *V* represents the volume, which can be defined as *V* = *Sh*, where *S* is the cross-sectional area and *h* is the layer thickness of the GeS_2_ monolayer. The layer thickness is obtained by the distance between the top and bottom surface atoms plus the Van der Waals radii of the surface atoms. $C_{\lambda }$ is capacity heat. At room temperature, the capacity heat follows the Dulong–Petit limit; thus, $\kappa_{l}$ is mainly contributed by $\nu_{\lambda }^{}$ and $\tau_{\lambda }$. Figure 9b,c show $\nu_{\lambda }^{}$ and $\tau_{\lambda }$ of the GeS_2_ monolayer under different tensile strains, respectively. Both $\nu_{\lambda }^{}$ and $\tau_{\lambda }$ decrease with an increasing tensile strain. This phenomenon leads to a decrease in the $\kappa_{l}$ with an increasing tensile strain, which agrees with our previous results. Moreover, the calculated average value of $\nu_{\lambda }^{}$ is reduced from 1.14 to 1.08 Km/s, while the average value of $\tau_{\lambda }$ decreases from 0.94 to 0.25 ps when the strain rises from 0 to 6%. Such small $\nu_{\lambda }^{}$ and $\tau_{\lambda }$ further guarantee the low $\kappa_{l}$ of the GeS_2_ monolayer. Furthermore, we also calculated the Grüneisen parameters of the GeS_2_ monolayer, as shown in Figure 9d. Interestingly, when the strain rises to 6%, the average value of Grüneisen parameters is enhanced from 1.11 to 3.15, indicating that anharmonic phonon interaction of the GeS_2_ monolayer is strengthened under tensile strain.

### 3.4. Thermoelectric Performance

Due to the relaxation time approximation in Boltzmann transport theory, we calculated the electron relaxation time before evaluating the quality factor ZT of the GeS_2_ monolayer. The carrier relaxation time can be defined as:(3)$$\tau =\frac{\mu m*}{e}$$
where the *μ* is carrier mobility, which can be estimated through deformation potential theory [55,56]:(4)$$\mu =\frac{2e\hbar^{3}C_{2D}}{3k_{B}T{\left|m*\right|}^{2}E_{i}^{2}}$$
where *e*, ℏ, k_B_, *T*, and *m** stand for the electron charge, reduced Planck constant, Boltzmann constant, temperature, and electron (hole) effective mass, respectively. The effective mass can be defined by: *m** = *ħ*^2^/(∂^2^*E*/∂*k*^2^), where *ħ* is the reduced Planck constant and *E* is the energy of the electron (hole) at wavevector *k* in the band. Therefore, the electron effective mass can be obtained from the second-order derivatives of the energy band near the conduction band minimum, while the hole’s effective mass is obtained from the energy band near the valence band maximum, and the corresponding fitting parameters are shown in Appendix A. $C_{2D}$ and $E_{i}^{}$ are the elastic modulus and deformation potential constant for 2D systems, respectively. Here, *C*_2D_ = 2(∂^2^(*E* − *E*_0_)/∂*ε*^2^)/*S*, where *S* is the cross-sectional area. Herein, the orthorhombic lattice of the GeS_2_ monolayer was built for the carrier mobility calculation, as plotted in Figure 10a. The band structure, total energy, and *E*_edge_ vs. strain for GeS_2_ monolayer in the orthorhombic unit cell are illustrated in Figure 10b–d, respectively. The corresponding parameters calculated and mentioned above are summarized in Table 1.

Finally, based on the thermoelectric parameters we obtained, the figure of merit ZT of the GeS_2_ monolayer under different tensile strains is plotted in Figure 11. Additionally, the figure of merit ZT as a function of carrier concentrations is also shown in Appendix A. Clearly, the tensile strain greatly enhances the ZT value of the GeS_2_ monolayer. The optimal ZT value at 300 K is 0.74 under a 6% strain, which is twice the strain-free GeS_2_ monolayer (ZT = 0.37). This phenomenon is mainly because the tensile strain enhances the PF while reducing both $\kappa_{l}$ and $\kappa_{e}$. More importantly, the ZT value will be increased from 0.74 to 0.92 with temperature increases from 300 to 700K. This value is comparable with the SiP_2_ monolayer (0.9 at 700 K) [57], TiS_2_ monolayer (0.95 at 300 K and an 8% tensile strain) [58], and WSSe monolayer (1.08 at 1500K and a 6% compressive strain) [48].

## 4. Conclusions 

In summary, by employing DFT calculations combined with semi-classical Boltzmann transport theory, the influence of tensile strain on the thermoelectric properties of the GeS_2_ monolayer was theoretically studied. Our findings manifest that the GeS_2_ monolayer exhibits indirect band gap semiconductor characteristics, and the band gap gradually increases with tensile strain. Moreover, the electronic and thermal transport properties of the GeS_2_ monolayer can be efficiently tuned by tensile strain. The tensile strain can significantly enhance the power factor while decreasing thermal conductivity, leading to the enhancement of the ZT value of the GeS_2_ monolayer. The lattice thermal conductivity of the GeS_2_ monolayer at 300 K is only 0.48 W/mK under 6% tensile strain. This phenomenon is mainly attributed to the ultralow phonon group velocities and phonon relaxation times of GeS_2_ monolayer under 6% strain. More importantly, the optimal ZT value of the 6% strained GeS_2_ monolayer at room temperature is about twice more significant than the case without strain. Our results give a new insight into the strain-modulated thermoelectric performance of the GeS_2_ monolayer.

## Figures and Tables

**Figure 1 materials-15-04016-f001:**
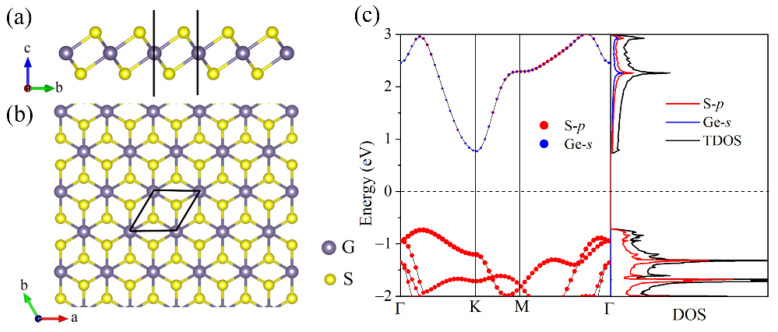
The structure of GeS_2_ monolayer’s (**a**) side and (**b**) top views. (**c**) The atom orbitals’ project band structure and DOS for GeS_2_ monolayer.

**Figure 2 materials-15-04016-f002:**
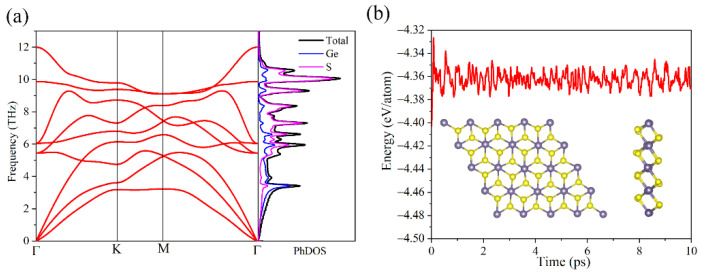
The (**a**) phonon spectrum and PhDOS of GeS_2_ monolayer and (**b**) total energies evolution and structure snapshots after 10 ps AIMD simulations at 300 K for GeS_2_ monolayer.

**Figure 3 materials-15-04016-f003:**
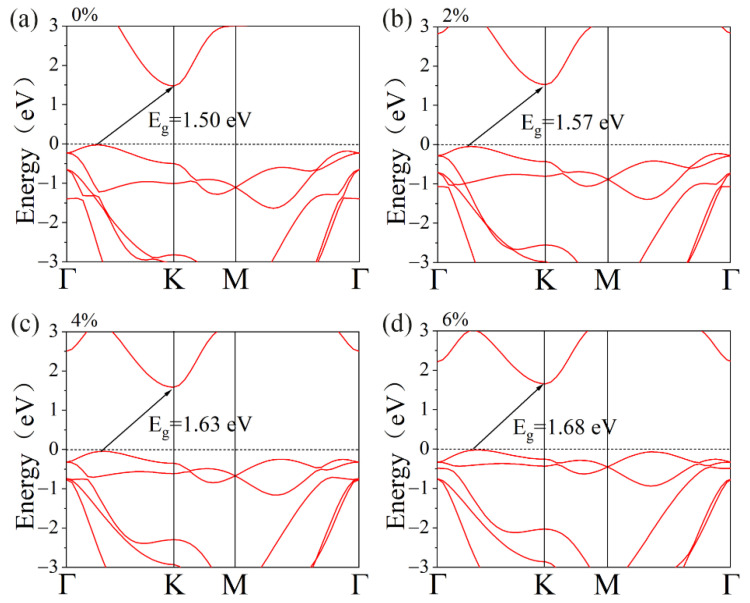
Band structures under different biaxial tensile strains of(**a**) 0%, (**b**) 2%, (**c**) 4% and (**d**) 6% for GeS_2_ monolayer.

**Figure 4 materials-15-04016-f004:**
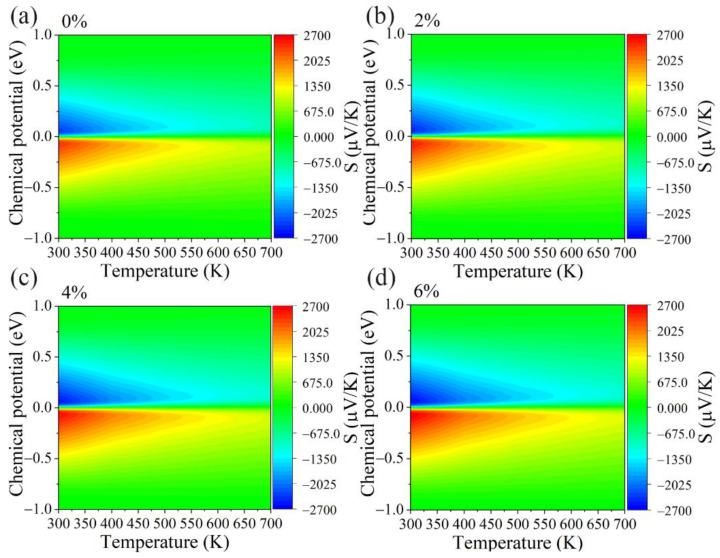
The contour maps of the Seebeck coefficient *S* with respect to chemical potential under different biaxial tensile strains of (**a**) 0%, (**b**) 2%, (**c**) 4% and (**d**) 6% for GeS_2_ monolayer.

**Figure 5 materials-15-04016-f005:**
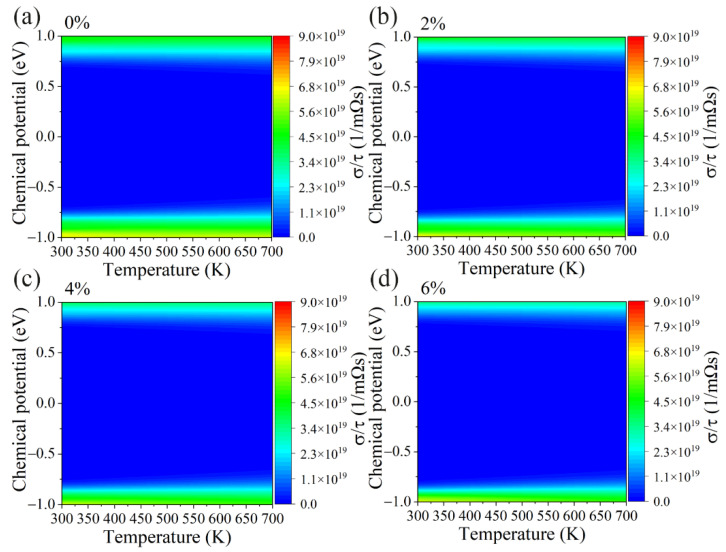
The contour map of the electrical conductivity divided by relaxation time (σ/τ) with respect to chemical potential under different biaxial tensile strains of (**a**) 0%, (**b**) 2%, (**c**) 4% and (**d**) 6% for GeS_2_ monolayer.

**Figure 6 materials-15-04016-f006:**
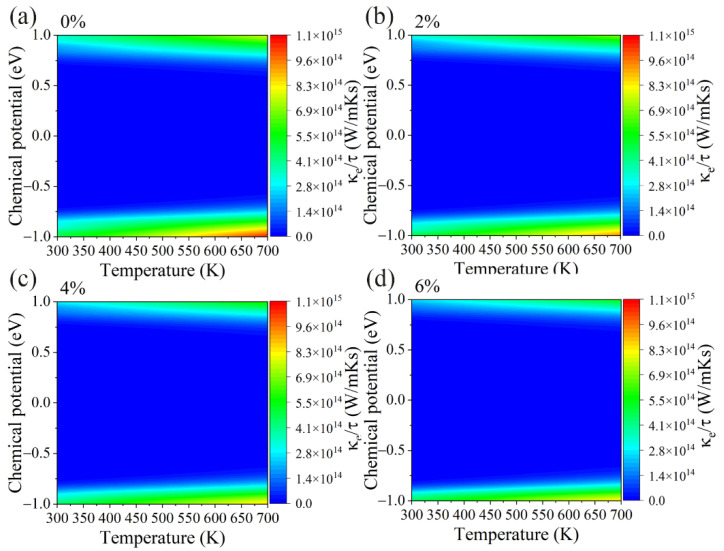
The contour map of the electronic thermal conductivity divided by relaxation time (κ_e_/τ) with respect to chemical potential under different biaxial tensile strains of (**a**) 0%, (**b**) 2%, (**c**) 4% and (**d**) 6% for GeS_2_ monolayer.

**Figure 7 materials-15-04016-f007:**
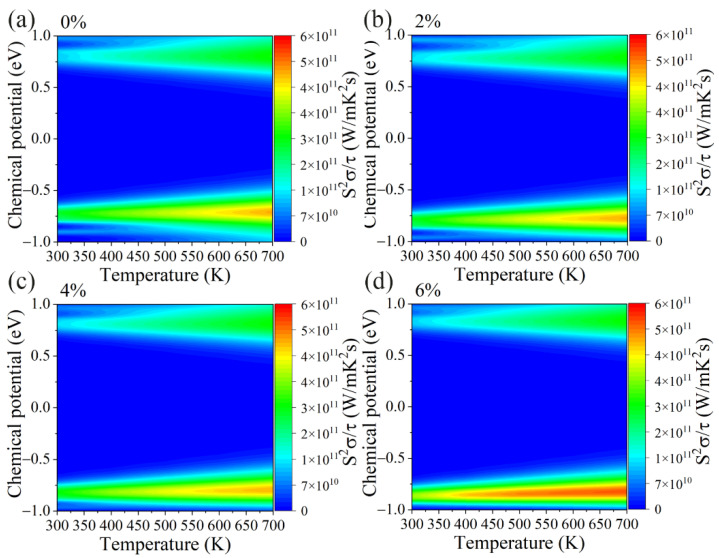
The contour map of the power factor divided by relaxation time (S^2^σ/τ) with respect to chemical potential under different biaxial tensile strains of (**a**) 0%, (**b**) 2%, (**c**) 4% and (**d**) 6% for GeS_2_ monolayer.

**Figure 8 materials-15-04016-f008:**
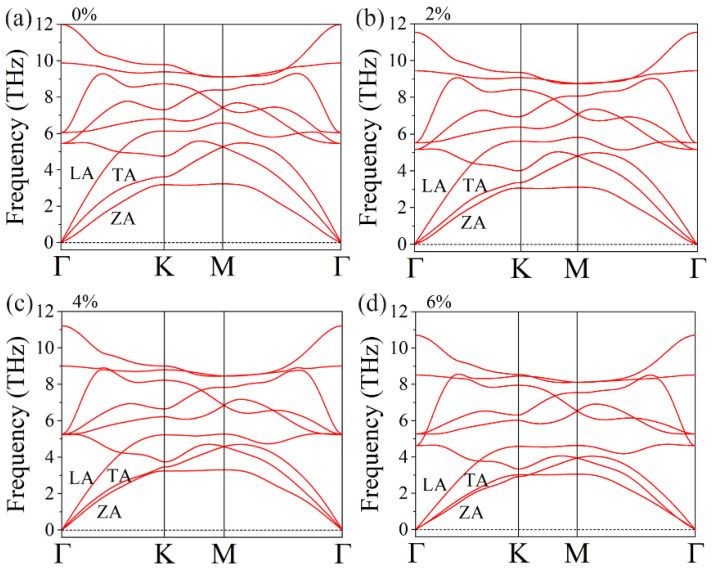
The phonon spectrum of GeS_2_ monolayer under different biaxial tensile strains of (**a**) 0%, (**b**) 2%, (**c**) 4% and (**d**) 6%.

**Figure 9 materials-15-04016-f009:**
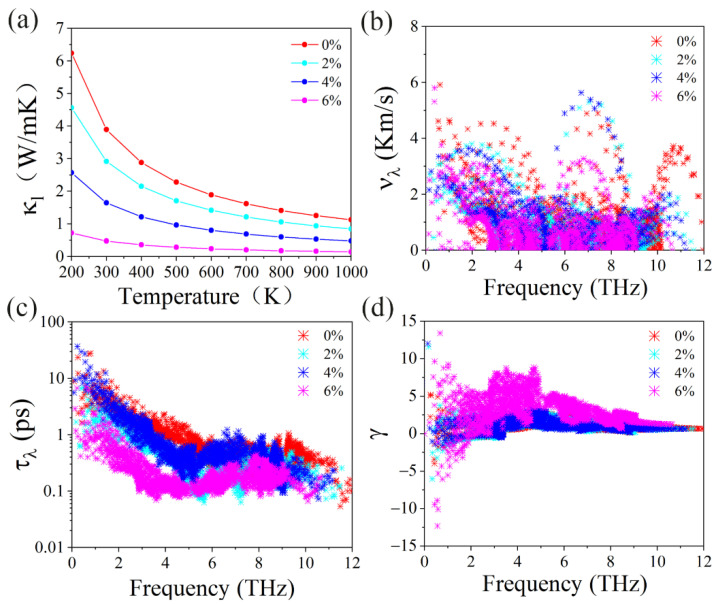
(**a**) The calculated lattice thermal conductivity $\kappa_{l}$ with respect to temperature under different tensile strains for GeS_2_ monolayer. The (**b**) phonon group velocity, (**c**) phonon relaxation time, and (**d**) Grüneisen constants for GeS_2_ monolayer at different tensile strains.

**Figure 10 materials-15-04016-f010:**
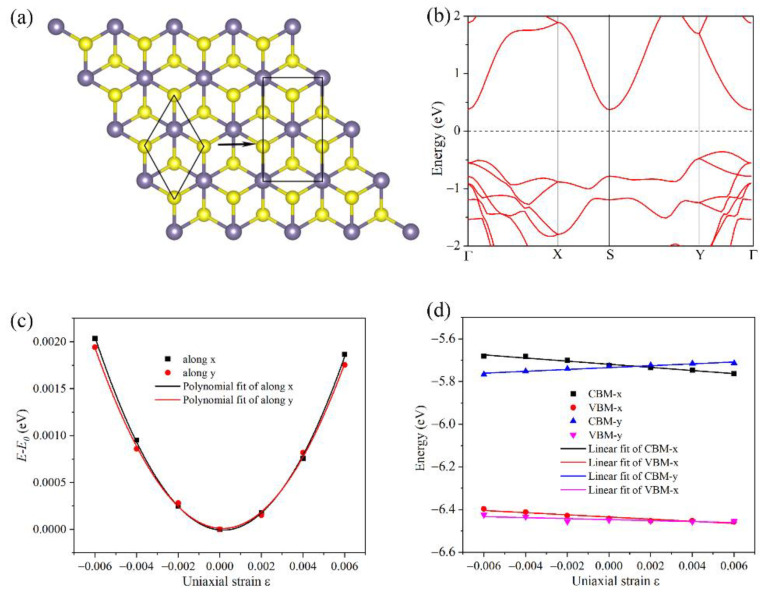
(**a**) The orthorhombic lattice of GeS_2_ monolayer. The calculated (**b**) electronic band structure, (**c**) total energy shift, and (**d**) band alignment for orthorhombic lattice GeS_2_ monolayer with respect to the uniaxial strain *ε* by PBE functional.

**Figure 11 materials-15-04016-f011:**
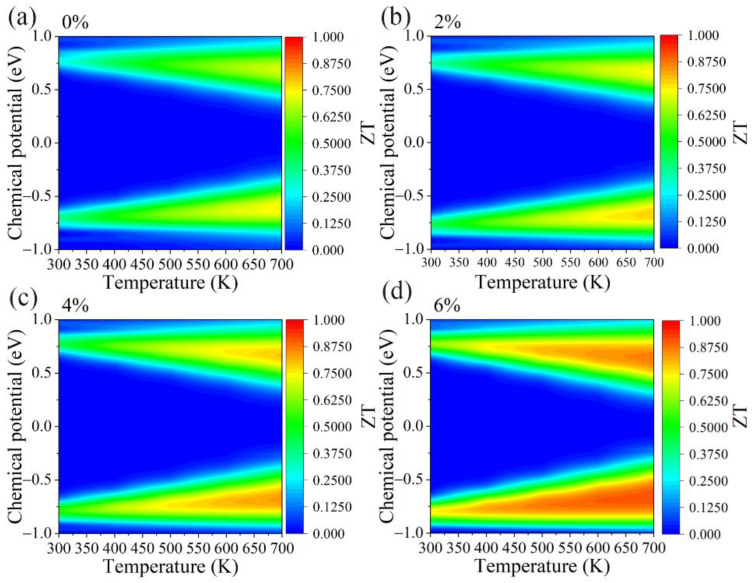
The contour map of the figure of merit ZT with respect to chemical potential under different biaxial tensile strains of (**a**) 0%, (**b**) 2%, (**c**) 4% and (**d**) 6% for GeS_2_ monolayer.

**Table 1 materials-15-04016-t001:** Calculated deformation potentials (*E*_l_), effective mass (*m**), elastic modulid (*C*_2D_), carrier mobility (*μ*), and electronic relaxation time (*τ*) of GeS_2_ monolayer under different directions.

Direction	Carrier Type	*E*_1_ (eV)	*C*_2D_ (N m^−1^)	*m*^*^/*m*_0_	*μ* (cm^2^ V^−1^ s^−1^)	*τ* (ps)
x	e	7.310	52.9	0.21	321.52	0.04
	h	5.065	52.9	0.88	37.41	0.02
y	e	4.359	49.9	0.68	79.80	0.03
	h	2.302	49.9	1.19	93.43	0.07

## Data Availability

The data presented in this study are available on request from the corresponding authors.

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
