# Peer review of "Strain-Enhanced Thermoelectric Performance in GeS2 Monolayer"

_materials, 2022, doi:10.3390/ma15114016_

Round 1

Reviewer 1 Report

In this work, the authors have studied the strain effect on the electronic and thermoelectric properties of 1T-GeS2 monolayer using first-principles calculations in combined with Semi-classical Boltzmann transport theory. 

The results seem to be very interesting and worth to be published in this journal. Nevertheless, there are some points needing to be clarified before the acceptance of this work.

1-   In Figure 2 (a), it is possible to add the PhDOS for pristine GeS2?

2-   The authors used the high-symmetry points -K-M- to sample the Brillouin zone for the band structure and phonon dispersion of pristine GeS2 monolayer and  -X-M- under biaxial tensile strain, I think should be uniformed for the entire article .

3-             The authors indicate that under biaxial tensile strain the valence band can reach maximum degeneracy, but we can notice that the valence bands between M and  point still move toward the Fermi level. For that, the authors should must explain why they chose to stop at 6 % value of tensile strain. What's the effect of compressive strain on the thermoelectric properties ?

4-      There are many writing errors in the text:

In the titles of Figures 2 and 3, GaS2 should be changed to GeS2.

5-             The manuscript should be written properly (mathematical expressions, interline…), and the English language requires improvements.

In summary, this work can be published provided that the authors give satisfying responses.

Reviewer 2 Report

The authors have well studies the GeS2 monolayer and its strain engineering for thermoelectric properties. I would like the paper be accepted as it is with few literary updates about 2D thermoelectrics e.g https://doi.org/10.1088/2516-1075/ac635b

Reviewer 3 Report

Please see tha attached report. 

Reviewer 4 Report

In this research, the authors study The strain effect on electronic and thermoelectric properties of GeS2 monolayer.

1.      For the electronic properties, Generalized Gradient Approximation is used by the authors, why not hybrid functionals are used?

2.      In the previously available data it is been reported that the GeS2 has an unsatisfactory ZT value of 0.23. So why do the authors consider GeS2 for further calculations?

3.      Effect of tensile strain on thermoelectric properties is reported, investigation of compressive strain is suggested to the authors.

4.       We ask the authors to carefully check the whole manuscript for spelling mistakes and grammar mistakes.

Reviewer 5 Report

This theoretical paper on the thermoelectric properties of strained germanium disulfide contains interesting ideas. Particularly, relating the thermoelectric figure of merit to the strain can be potentially useful to practical device applications. The visualization of the numerical data is also appealing. However, I find some important details in the methods are missing making a full assessment of the available data difficult. Therefore, the paper must be revised to include this information and reassessed again:

The authors should provide a justification for the suitability of the functional used either by conducting comparative simulation or quoting successful similar cases from the literature. Also, the description of functionals used in the methods section is confusing; Which simulations were exactly performed with BPE and which with the hybrid functional? For the hybrid functional, what were the screening range and the mixing parameter?

For the molecular dynamics, the authors should provide the ensemble used and the time step parameter.

The procedure for calculating the effective masses should be explicitly elaborated on. What bands were used for calculating the effective masses? Where were the band extrema points located? How was the numerical differentiation performed (especially the delta k value)? Also, the effective mass components should be given as well.

Finally, the significance of many of the questions raised above are described in detail in the following references that authors can consult with and probably quote in their paper:

Moreno et al. A review of recent progress in thermoelectric materials through computational methods, Materials for Renewable and Sustainable Energy volume 9, Article number: 16 (2020); https://doi.org/10.1007/s40243-020-00175-5

Mbaye et al. Data-driven thermoelectric modeling: Current challenges and prospects, Journal of Applied Physics 130, 190902 (2021); https://doi.org/10.1063/5.0054532

Round 2

Reviewer 3 Report

This manuscript can now be published in the present form.

Reviewer 5 Report

The authors have addressed my concerns. The paper can be recommended for publication.